# Fungal Endophytes: Discovering What Lies within Some of Canada’s Oldest and Most Resilient Grapevines

**DOI:** 10.3390/jof10020105

**Published:** 2024-01-26

**Authors:** Shawkat Ali, A. Harrison Wright, Joey B. Tanney, Justin B. Renaud, Mark W. Sumarah

**Affiliations:** 1Agriculture and Agri-Food Canada, Kentville Research and Development Centre, 32 Main St., Kentville, NS B4N 1J5, Canada; harrison.wright@agr.gc.ca; 2Natural Resources Canada, Pacific Forestry Centre, 506 Burnside Road West, Victoria, BC V8Z 1M5, Canada; joey.tanney@nrcan-rncan.gc.ca; 3Agriculture and Agri-Food Canada, London Research and Development Centre, 1391 Sandford St., London, ON N5V 4T3, Canada; justin.renaud@agr.gc.ca (J.B.R.); mark.sumarah@agr.gc.ca (M.W.S.)

**Keywords:** *Botrytis*, fungal endophytes, grapevines, *Diaporthe*, small metabolites

## Abstract

Plant diseases and pests reduce crop yields, accounting for global crop losses of 30% to 50%. In conventional agricultural production systems, these losses are typically controlled by applying chemical pesticides. However, public pressure is mounting to curtail agrochemical use. In this context, employing beneficial endophytic microorganisms is an increasingly attractive alternative to the use of conventional chemical pesticides in agriculture. A multitude of fungal endophytes are naturally present in plants, producing enzymes, small peptides, and secondary metabolites due to their bioactivity, which can protect hosts from pathogens, pests, and abiotic stresses. The use of beneficial endophytic microorganisms in agriculture is an increasingly attractive alternative to conventional pesticides. The aim of this study was to characterize fungal endophytes isolated from apparently healthy, feral wine grapes in eastern Canada that have grown without agrochemical inputs for decades. Host plants ranged from unknown seedlings to long-lost cultivars not widely propagated since the 1800s. HPLC-MS was used to identify unique endophyte-derived chemical compounds in the host plants, while dual-culture competition assays showed a range in endophytes’ ability to suppress the mycelial growth of *Botrytis*, which is typically controlled in viticulture with pesticides. Twelve of the most promising fungal endophytes isolated were identified using multilocus sequencing and morphology, while DNA barcoding was employed to identify some of their host vines. These fungal endophyte isolates, which consisted of both known and putative novel strains, belonged to seven genera in six families and five orders of Ascomycota. Exploring the fungal endophytes in these specimens may yield clues to the vines’ survival and lead to the discovery of novel biocontrol agents.

## 1. Introduction

When Leif Eriksson and his crew of Icelandic Norsemen first travelled to North America (c. 1000 AD), they called it *Vinland*, purportedly for the wild grapes that grew in abundance there [1,2]. The people indigenous to what is now referred to as the Canadian Maritimes were of course familiar with these wild grapes, which, in the form of fresh fruit, made up a small, seasonal part of their diet [3]. When a second wave of both English and French European explorers and settlers began to arrive in the early 1600s, several contemporary accounts note their interest in the wild *Vitis* species that they encountered [4]. Furthermore, a handful of optimistic European settlers, possibly inspired by their observations of the local flora, established and tended early small-scale plantings in the Canadian Maritimes, including some of the first vines to be cultivated in Canada [4]. 

Although favourable microclimates for grape growing exist in the Maritimes, and the local grape industry is currently undergoing rapid expansion and growth [5], likely due in part to recent climate warming, the region’s climate does not immediately seem auspicious for viticulture. In fact, commercial viticulture in the region was deemed untenable by early researchers [6]. While the maritime influence results in relatively mild winters, the region also experiences high humidity and heavy precipitation and has a shorter growing season and fewer growing degree days (GDDs) than other important grape-growing regions in Canada. In spite of this, a genuine wine grape industry was born and now thrives there, with the first commercial vineyards springing up in the 1970s and 1980s [4]. However, remnants of much earlier plantings—with both North American and European lineages, some dating back centuries—still persist in many places throughout the Maritimes today. So, given the long history of both indigenous and European grapes in the Maritimes, which is on the edge of where wine grapes are traditionally grown, what microbial communities might be living inside some of these specimens that have stood the test of time?

Complex and diverse microbial communities (microbiomes) exist inside all plants. Although the relationship between these microbes and their hosts ranges from mutualism to pathogenicity, most of these associations are in the little-known middle portion of this spectrum, according to the current understanding [7,8]. Some of these relationships benefit both the endophyte and its plant host, and possibly co-evolved over time [9]. The means by which these endophytes may improve plant health or defend a host plant from pathogens can take various forms, such as competition for space, the production of bioactive metabolites, including antifungal agents, mycoparasitism, or the promotion of plant growth via hormones or an upregulation of natural defense mechanisms [10,11,12,13]. 

Unfortunately, for the vast majority of the history of modern horticulture, the relationship between the plants we grow and these microorganisms has been largely ignored. Instead, gains in productivity have been achieved by fulfilling plants’ growth requirements using synthetic fertilizers and eliminating factors harmful to growth, such as pests and disease, which account for global losses in the range of 30% to 50%, through the use of pesticides. This over-simplified approach ignores and often disrupts the beneficial natural roles played by many microorganisms. In recent times, with a greater understanding of the adverse effects of agrochemicals on the environment and animal and human health and the resulting tightening of residue restrictions, public pressure is mounting to reduce the use of synthetic fertilizers and pesticides in agriculture. This change is encouraging both governments and private companies to pursue cleaner alternative technologies, including the use of beneficial endophytes as potential growth promoters and biological control agents in commercial plant production [9,14,15,16,17,18].

While other research groups have previously explored the use of endophytic fungi in wild grapes as potential biocontrol agents [19,20], our study is unique in its focus on wine grape cultivars growing ferally in a region probably quite unlike that where most of their evolution took place. The main objective of this study was to isolate, identify, and characterize the fungal endophytes in vines (especially those with some European ancestry) that have managed to persevere and even thrive in the cool**,** wet climate of the Maritimes for many decades or even centuries. We, therefore, isolated fungal endophytes from the leaves of the oldest vines and identified them through multilocus sequencing and morphology. We also investigated the chemicals that these endophytes produce in order to understand their potential roles as biocontrol agents. While the focus of this research was endophyte pre-screening, exploring the fungal endophytes found in these specimens could yield clues about their resiliency, as well as lead to the discovery of novel growth-promoting and biocontrol agents as part of future research.

## 2. Materials and Methods

### 2.1. Collection of Grape Samples and Varietal Testing

More than 80 leaf samples were collected from grapevines across the province of Nova Scotia, Canada. Candidates chosen for sampling were unsprayed, uncared for, and relatively free from disease. Promising specimens were found in locations ranging from woods and the edge of meadows (often long-lost farmsteads) hosting wild and feral grapevines to historic properties containing unruly patches of grapevines of unknown origin. Varietal testing was performed using DNA barcoding (Foundation Plant Services, UC Davis, CA, USA) on vines from a small subsection of the locations thought to be the most likely to contain early named cultivars (e.g., specimens growing near early European settlement sites). Four Canadian sites, and the vines found on them, were chosen for DNA barcoding. They included (1) feral vines in the woods surrounding Bear River, Nova Scotia, Canada a town founded in the early 1600s and rumoured to have early grape plantings; (2) a vine located in the Annapolis Royal Historic Gardens (Annapolis Royal, NS) sourced from feral vines in Bear River, Nova Scotia, and transplanted to its present location in 1980; (3) a vine growing around the old foundations of a former farmstead in Miller Point Peace Park outside Lunenburg, Nova Scotia, a UNESCO World Heritage Site. This vine was supposedly a ‘Diana’ and approximately 170 years old, according to a nonagenarian living in the area who said that, as a young boy, he had harvested fruit from it, as had his father and grandfather (personal communication), and (4) the French Mission in Poplar Grove, Nova Scotia, purportedly built in 1699 and believed to be the oldest standing building in Canada east of Quebec. Grapevines grow all over this historic property and locals believe that early monks used the grapes at this site to make communal wine.

### 2.2. Leaf Sterilization and Isolation of Foliar Fungal Endophytes

The sterilization and isolation procedure was modified from that described in [21]. Five healthy leaves were randomly chosen from each vine sampled. For each leaf sample, the petiole was removed, and the blade was bisected by a transverse cut. Three 1 cm^2^ pieces were then cut out of the basal portion of each leaf and retained; the remainder of the leaf was discarded. The five petioles and 15 leaf pieces, representing one sample, were placed inside a tea infuser. In a sterile laminar flow hood, the infuser containing the sample was immersed in a beaker containing 75% ethanol, swirled for one minute, and then transferred to a beaker containing 1 L of 6% sodium hypochlorite solution and 0.05% TWEEN 80, which was swirled intermittently for 7.5 min. The infuser was then transferred to a second ethanol beaker and swirled for an additional minute before being rinsed in a series of three beakers containing sterile ultrapure water (Barnstead™ Nanopure™ D11971, Van Nuys, CA, USA). Without removing them from the sterile environment, the leaf pieces were transferred to a sterilized paper towel. Interior ~0.5 cm^2^ pieces were removed from each section; the two ends were cut off each petiole, and the remaining piece was cut into three equal pieces of approximately ~0.5 to 1 cm in length, making 30 pieces per sample. To test sterilization efficacy, three pieces of the petiole and three pieces of the blade were pressed onto plates containing half-strength potato dextrose agar (PDA) for five minutes on each side; the plates were then sealed with parafilm and checked after two weeks to confirm the absence of microbial growth. For each sample, three pieces of each tissue type were then transferred to a plate of half-strength PDA for a total of five plates of each tissue type. The plates were then sealed with parafilm and stored in the dark at room temperature. The cut ends and edges were checked for endophyte growth every few days. To obtain pure fungal isolates, the mycelium growing out from these pieces was transferred to fresh PDA plates; in some cases, a second or third sub-culturing of the isolates was performed. For the purpose of this study, we focused exclusively on slow-growing fungal endophytes, i.e., those that took at least six days to emerge from the plated plant tissue sample. The isolated endophytes were given a sample ID based on the order in which the site/field sample was collected and the order in which the endophyte emerged from the sample after a minimum of six days (e.g., En01-2 would be the second endophyte to emerge, after a minimum of 6 d, from the first site/sample collected). For samples that produced only one endophyte, the second number was not used (e.g., En60).

### 2.3. Culturing and Morphological Observations

To induce in vitro sporulation, strains were inoculated on several media, including 2% malt extract (MEA) (20 g Bacto malt extract, Difco Laboratories, Sparks, MD, USA; 15 g agar, EMD Chemicals Inc., Gibbstown, NJ, USA; 1 L distilled water); cornmeal agar (CA) (Difco Laboratories, Detroit, MI, USA); potato dextrose agar (PDA) (Difco Laboratories, Detroit, MI, USA); oatmeal agar (OA) (extract of 30 g/L boiled oatmeal, 15 g agar, 1 L distilled water); V8 juice agar (V8A) (200 mL Campbell’s V8 juice, 15 g agar, 2 g CaCO_3_, 800 mL distilled water); and water agar (WA) (20 g agar, 1 L tap water), with or without the addition of sterile filter paper or autoclaved leaves of *Gaultheria shallon*, *Hedera helix*, *Ilex aquifolium*, or *Rubus armeniacus* [22]. Cultures were incubated at 20 °C in the dark or with a 12 h:12 h fluorescent light cycle, or near a window under ambient conditions.

Micromorphological characters were visualized, described, and measured from living material mounted in deionized or tap water using a Leica DM4 B light microscope (Leica Microsystems CMS GmbH, Wetzlar, Germany). Hand sections of perithecia and pycnidia were obtained using a double-edge safety razor blade. Micrographs were captured using a Leica DFC450 C camera (Leica Microsystems CMS GmbH, Wetzlar, Germany) with Leica LAS X 33.7.1 software. Colonies were observed and photographed using a Leica M165 C stereomicroscope with a TL5000 Ergo light base (Leica Microsystems Ltd., Singapore) and a Leica DMC5400 camera. Photographic plates were assembled using Adobe Photoshop CC 2019 v.23.0.1 (Adobe Systems, San Jose, CA, USA).

### 2.4. DNA Extraction, Amplification, Sequencing, and Analysis

For DNA extraction, a sterilized scalpel was used to scrape 10–15 mg of mycelium from 5–10-day-old cultures grown on PDA medium at 22 °C. The mycelium was then macerated with steel beads in a 2 mL Eppendorf tube using a Qiagen TissueLyser II (Qiagen, Hilden, Germany) for 2 × 60 s cycles at 30 beats per second. Total genomic DNA was isolated using an E.Z.N.A.^®^ SP Fungal DNA Mini Kit (Omega Bio-Tek, Norcross, GA, USA), following the manufacturer’s instructions. The partial β-tubulin (TUB2), internal transcribed spacer (ITS), translation elongation factor 1-alpha (TEF-1α), RNA polymerase second largest subunit (RPB2), partial actin (ACT), chitin synthase I (CHS-1), and LSU genes were amplified by PCR using the primer pairs shown in Appendix A. The PCR amplification was carried out in a 50 μL reaction mixture using the same PCR conditions as previously described [23,24]. PCR-amplified products for all genes were run on 1% agarose in a 1X Tris-borate-EDTA running buffer to confirm the size and amplification of the single band. The PCR-amplified products were then cleaned with an ExoSAP-IT™ PCR Product Cleanup kit (Applied Biosystems, Waltham, MA, USA), following the manufacturer’s protocol. Sequencing was carried out at the Genome Quebec Innovation Centre and Eurofins Genomics using ABI BigDye 3 Terminator Cycle sequencing chemistry (Applied Biosystem’s 3730xl DNA Analyzer technology).

Sequence contigs were assembled and trimmed using Geneious Prime 2019 v.2019.0.4 (Biomatters, Auckland, New Zealand). Using BLASTn (Nucleotide BLAST: Search nucleotide databases using a nucleotide query (nih.gov), https://blast.ncbi.nlm.nih.gov/Blast.cgi?PROGRAM=blastn&BLAST_SPEC=GeoBlast&PAGE_TYPE=BlastSearch, accessed on 10 August 2023), the ITS sequences obtained were compared with those in the National Centre for Biotechnology Information (NCBI) GenBank database, and strains were tentatively identified against available ITS sequences in the NCBI GenBank. Following these identifications, additional taxon-specific secondary barcodes were sequenced and analyzed (Table 1 and Appendix A). Phylogenetic analyses were performed for each taxon using the appropriate gene sequence datasets populated with sequences based on NCBI BLAST queries. Sequences were aligned using MAFFT (Multiple Alignment using Fast Fourier Transform, https://mafft.cbrc.jp/alignment/server/index.html, accessed on 10 August 2023) [25] and then manually verified and adjusted if required. Phylogenetic trees were constructed with maximum likelihood (ML) using IQ-TREE v1.6.11 [26]. The best model for each partition was selected using ModelFinder [27], which performed 1000 ultrafast bootstraps [28], 1000 SH-aLRT branch tests, and an approximate Bayes test [29]. The best-fit substitution models based on the Bayesian information criterion (BIC), selected outgroup taxa, and other information on each analysis are presented in Table 1. Phylogenetic trees were built with Geneious (Geneious Prime 2019 v.2019.0.4 (Biomatters, Auckland, New Zealand) and edited with Adobe Illustrator CC 2019 v.23.0.1.

### 2.5. Screening of Endophyte Bioactivity against Botrytis

The isolated fungal endophyte strains were screened for their bioactivity against the common grape pathogen *Botrytis cinerea* using dual-culture competition assays [10]. Briefly, a 4 mm agar plug containing mycelia was removed from both a freshly grown pure culture of *Botrytis cinerea* and the endophyte isolate of interest and transferred to a new 9 cm PDA plate using an aseptic technique. The plugs were kept 7 cm apart from each other and 1 cm away from the edge of the plate. The starting time of the co-culture was based on the growth rate of endophytes and pathogens. In instances where the endophyte grew slowly compared to *Botrytis*, the endophyte was incubated on the plate for 3–5 days before the pathogen was added; before the endophyte reached the middle of the plate, the same plate was inoculated with *Botrytis*. The experiment was carried out in triplicate and repeated three times. As a control, a solitary plug of either *Botrytis cinerea* or the endophyte strain was placed on PDA plates to ensure the pathogen grew normally in the absence of the endophytes, and the endophyte grew normally in the absence of *Botrytis*. The plates were incubated in the dark at 22 °C and the growth of both endophyte and pathogen were observed daily. Before the two cultures touched each other, the growth of *Botrytis* on each plate was measured towards the pathogen (P) and was oriented vertically (VP) and oriented horizontally towards both sides of the plate (HP). The percent growth inhibition index (GII) was calculated as (HP-VP)/HP × 100 [76]. Only endophytes that showed some ability to inhibit *Botrytis* growth (GII > 5% based on averaging alone) were characterized further. An ANOVA was later performed on this group to test for significance, while a post hoc Tukey multiple means comparison was used to determine if specific averages differed significantly [77].

### 2.6. Metabolite Screening by High-Resolution HPLC-MS

Extracts from fungal endophytes and *Botrytis* were analyzed using a Q-Exactive Quadrupole-Orbitrap mass spectrometer (MS) (Thermo Scientific, Waltham, MA, USA) coupled with an Agilent 1290 ultra-high-performance liquid chromatography (HPLC) system. Metabolites were resolved using a Zorbax Eclipse Plus RRHD C18 column (2.1 × 50 mm, 1.8 μm; Agilent Technologies, Santa Clara, CA, USA) maintained at 35 °C. The mobile phase consisted of (A) water with 0.1% formic acid and (B) acetonitrile with 0.1% formic acid (Optima grade, Fisher Scientific, Branchburg, NJ, USA). The gradient consisted of 0% B for 30 s before increasing to 100% over 3 min. The mobile phase B was held at 100% for 2 min before returning to 0% over 30 s. Heated electrospray ionization was performed in positive mode using the following settings: capillary voltage, 3.9 kV; capillary temperature, 400 °C; sheath gas, 19 units; auxiliary gas, 8 units; probe heater temperature, 450 °C; and S-Lens RF level, 45.00. MS data were acquired using untargeted data-dependent acquisition (DDA), which included a full MS scan at a 70,000 resolution with a scan range of 106.7–1600 *m/z*, an automatic gain control target of 3 × 10^6^, and a maximum injection time of 250 ms. The five most intense ions in each full scan were selected for tandem mass spectrometry (MS/MS) analysis using a 1.2-Da isolation window and were analyzed under the following conditions: resolution, 17,500; automatic gain control target, 1 × 10^6^; max injection time, 64 ms; normalized collision energy, 35%; intensity threshold, 1.5 × 10^5^; and dynamic exclusion, 5 s.

## 3. Results

### 3.1. Grape Site/DNA Barcoding Results

The DNA barcoding results for the grapevines sampled from the four most promising (i.e., historically significant) sites were as follows: (1) surrounding woods, Bear River, NS, Canada: most likely a seedling from an unknown mix of North American species and *V. vinifera* ancestry; (2) Annapolis Royal Botanical Gardens, Annapolis Royal, NS: ‘Isabella’; (3) abandoned farmstead, Miller Point Peace Park, Lunenburg, NS: a seedling made up of entirely native North American species; and (4) French Mission, Poplar Grove, NS: ‘Clinton.’

### 3.2. Isolation and Identification of Endophyte Strains

A total of 12 fungal endophyte strains met the criteria for further characterization: (a) they were collected from a historically significant site; (b) they did not emerge from the explant material for at least six days; and (c) they showed some antagonism towards *Botrytis* based on averaging (GII > 5%). On the basis of morphology, GenBank queries, and subsequent phylogenetic analyses, nine of the twelve strains were confidently identified as species (Table 1). Six strains belonged to the genus *Diaporthe* (Diaporthaceae, Diaporthales), including four strains of *D. eres* (formally *D. vacuae*; En25-6, En26-4, En26-5, En61-3), which make up part of a strongly supported clade containing the former *D. vacuae* (CAA823) and *D. cotoneastri* (CBS 439.82) types in the TUB2-EF1-ITS phylogeny (Figure 1). En20-4 was not conclusively identified and was placed basally in a strongly supported clade that includes the *D. angelicae* (CBS 111592), *D. cucurbitae* (DAOM 42078), and *D. gulyae* (BRIP 54025) types. The remaining strain (En01-1) is a putative novel species basal to the *D. gulyae* clade and a clade including the *D. arctii* (DP0492) and *D. neoarctii* (CBS 109490) types. The En25-6 (*D. eres*) strain formed condiomata that produced alpha conidia on water agar amended with *Ilex aquifolium* leaves (Figure 2A–D); En20-4 produced alpha and beta conidia on OA (Figure 2E–H); and En01-1 produced both alpha and gamma conidia and fertile perithecia on WA amended with *Rubus armeniacus* leaves (Figure 2I–N). En61-1 was identified as *Gnomoniopsis paraclavulata* (Diaporthaceae, Diaporthales) based on its sequence (ITS and RPB2) similarity with the type specimen (BPI 877448) and the morphology of the conidiomata and the conidia formed on WA amended with *R. armeniacus* leaves (Figure 3I–K and Figure 4C).

En25-3 is *Colletotrichum fioriniae* (Glomerellaceae, Glomerellales), with 100% similarity to the ITS and CHS-1 sequences and the ex-type specimen of *C. fioriniae* (CBS 128517) (Figure 5A). Conidiomata production was stimulated on WA with the addition of *R*. *armeniacus* leaves (Figure 3G–H). En61-2 was identified as *Neophaeomoniella niveniae* (Celotheliaceae, Phaeomoniellales) based on the similarity to the ITS and EF1a sequences with the ex-type *N. niveniae* strain (CBS131316) and the comparison of its conidiomata and yeast-like conidia, which were abundant when grown on V8A (Figure 3A–F and Figure 4B). En25-1 was identified as *Nemania aureolutea* (Xylariaceae, Xylariales;) based on its 100% similarity between its ITS sequence and the authenticated strain MAR101219 and by its orange colony, a character distinguishing *N. aureolutea* from the closely related *N. aenea* [61] (Figure 5B and Figure 6A).

En60 is an unidentified *Ramularia* sp. (Mycosphaerellaceae, Mycosphaerellales), which shares a 99% similarity with the ITS sequences of the ex-types of *R. heraclei* (CBS 108969) and *R. weberiana* (CBS 136.23) and authenticated cultures of *R. alangiicola* (CPC 10299), *R. gei* (CBS 113977), *R. interstitiales* (CBS 120.68), *R. ligustrina* (CBS 379.52), *R. pratensis* (CBS 122105), and *R. rumicicola* (CPC 11295). The concatenated *ACT*-*ITS-RPB2* phylogeny placed En60 in a strongly supported clade with the ex-types of *R. heraclei* (CBS 108969) and *R. lamii* var. *lamii* (CBS 108970) (Figure 4D and Figure 6B–F). En26-1 was identified as *Sphaerulina amelanchieris* (Mycosphaerellaceae, Mycosphaerellales) based on its close sequence similarity (99% *EF1*, ITS) to the ex-type strain CBS 135100 (Figure 4A and Figure 6G–H).

### 3.3. Screening of Antimicrobial Activity of Endophytic Fungi against Botrytis

According to the dual-culture competition assays performed on the twelve selected isolates thought to be potentially active against *Sh*, only five of the twelve isolates initially screened showed statistically significant antagonistic activity. Values ranged from a GII of 27% (En61-3) to 35% (En26-5), while no inhibition (GII = 0%) was observed in the control (Figure 7 and Appendix A). All five of the significantly antagonistic strains belonged to the genus *Diaporthe*, with four out of the five identified as *D. eres*.

### 3.4. Extrolite Screening of Bioactive strains

The full complement of ionizable compounds in the extracts was visualized using principal component analysis (PCA) (Appendix A). As expected, the four strains of *D. eres* were grouped together; however, no significant clusters of the other strains were observed. Specifically, the other *Diaporthe* species (corresponding to isolates En20-4 and En01-1) did not strongly group with *D. eres. C. fioriniae* (En25-3) was an outlier (Appendix A): the strain’s unique extrolite profile that is responsible for its outlier position on the PCA plot was driven by unknown compounds (Appendix A). Four of the extrolites, with the formulas C_18_H_34_O_5_, C_21_H_34_O_4_, C_11_H_20_O_6_, and C_11_H_18_O_5_, are likely structurally similar according to their respective MS/MS spectra (Appendix A). Searching through published microbial compound databases [78] did not yield any likely candidates for these compounds, thus making them attractive targets for isolation and characterization. Strain En25-3 also had higher levels of carnitine, a common quaternary ammonium compound, than the other tested strains.

En61-1, like *N. niveniae,* produced several known, but unique, secondary metabolites (Appendix A and Appendix A). A number of linear dipeptides were detected as products of this strain, notably Ser-Val, Leu-Val, Ile-Thr, Ile-Ile, Ile-Pro, and Ile-Phe, although the positions of the amino acids are only putative. The series of unique extrolites produced by the strain *G. paracalvulata* was limited compared to the other tested strains and showed some degree of overlap with *N. niveniae*. Two metabolites detected in this strain have the formulas C_20_H_21_NO_6_ and C_25_H_29_NO_8_, which, based on their respective MS/MS product spectra, are likely structurally similar (Appendix A). The similarity of extrolite production in *G. paracalvulata* and *N. niveniae* is shown by their close proximity in Appendix A.

Six of the strains tested for metabolite profiling were identified as belonging to the order Diaporthales (Figure 1 and Table 1). In agreement with the phylogeny established, the four strains identified as *D. eres* were found to produce a similar complement of metabolites and, therefore, are clustered together in the PCA plot (Appendix A). The En20-4 and En1-1 strains did not strongly group with each other, nor did the *D. eres* strains (Appendix A). Some metabolites were found to be common across most of the tested *Diaporthe* strains, notably the osmoprotectant proline betaine. Other metabolites detected in the *D. eres* strains include tryptamine and phenylethylamine. Additionally, a number of related metabolites detected in *D. eres* include C_18_H_35_NO_2_, C_18_H_33_NO_2_, C_18_H_35_NO, and C_18_H_33_NO (Appendix A).

## 4. Discussion

The biological control of plant diseases through the use of antagonistic endophytic microorganisms is an alternative approach for reducing or eliminating the use of chemical pesticides in agriculture. Fungal endophytes belong to a taxonomically and metabolically diverse group of organisms that colonize different plant niches without causing any harm to the host plants. Their efficacy in controlling various plant pathogens has been investigated by a number of authors [79,80,81,82].

### 4.1. Feral Endophyte Host Identification

The results of DNA barcoding showed that although some vines had purely North American ancestry, others were a mix of North American and European lineages. It is no surprise that some of the tissue samples sent for DNA barcoding were found to be from unnamed seedlings. Feral grapes growing in the woods or on the edges of fields and roadways have become commonplace in the region, and are likely natural crosses distributed by birds and animals. This is most likely the case for the vine from the woods surrounding Bear River, NS, which was shown to be a mix of *Vitis vinifera* and North American species. However, one unexpected finding was the discovery that the old vine located on a long-abandoned homestead outside the historic town of Lunenburg, NS—widely known for nearly two centuries and believed to be ‘Diana’—is actually unnamed and a product of exclusively native American species. Native North American species are not widespread in Nova Scotia, even though they are abundant in the neighbouring province of New Brunswick. Lastly, our search yielded two named heritage cultivars: ‘Clinton,’ found at the French Mission in Poplar Grove, NS, and ‘Isabelle,’ relocated to the Annapolis Royal Historic Gardens, Annapolis Royal, NS from a specimen found growing wild over 40 years ago in Bear River, NS. Both cultivars are of historical importance and have likely been growing in Nova Scotia for well over a century. ‘Clinton’ is purely North American in its lineage, and thought to be primarily derived from *Vitis riparia*, but with some *Vitis labrusca* traits. It came to prominence in the 1830s and is widely cited as the first cultivated grape of *Vitis riparia* lineage. It was popular in the 19th century because of its high vigour, hardiness, and fruitfulness, and its tolerance to Phylloxera. This cultivar was once a favourite in both North America and France but is now rarely found, although it continues to be influential due to its frequent use in the past by grape breeders and its role as a founding parent of many important cultivars [83]. ‘Isabella’ is the product of a natural cross between *Vitis labrusca* and *Vitis vinifera* that was later discovered and named. The cultivar was introduced in 1816 and was the mainstay of the grape industry for the next half-century in New England and the neighbouring North Atlantic region, with extensive plantings also found in Europe. However, the once-popular cultivar eventually fell out of favour. A factor in its downfall may have been the widely held belief that the source of the Great French Wine Blight, a phenomenon that devastated the French wine industry, was Phylloxera brought from North America to Europe on the roots of the ‘Isabella’ vine [83].

### 4.2. Botrytis Suppression

In this study, five strains of *Diaporthe* successfully inhibited the growth of *Botrytis* in dual-culture competition assays (Figure 7). *Botrytis* is a major grapevine pathogen, reducing yield through bunch rot and premature cluster drop, and lowering table grape quality through postharvest fruit rot. Several spray applications of a chemical fungicide are typically used annually to control *Botrytis* rot, which not only increases production costs but also raises a range of the environmental issues associated with agrochemical use. Although some biological control agents have been registered for the management of *Botrytis* rot in grapevines, they are not very effective in completely eliminating the disease. It has been reported that species of the genus *Diaporthe* can alter the lifestyle between phytopathogens and endophytes on the same host or between different hosts [84,85].

In future work, it would be interesting to test *Diaporthe* isolates from this study in the *in planta* inoculation of grapevine and then challenge half of the plants with *Botrytis* and other common grapevine pathogens. Plants inoculated with *Diaporthe* but not challenged with pathogens will show whether these strains are pathogenic to grapevines. *D. eres* isolated from *Prunus dulcis* has been reported to have significant antifungal activity against three fungal pathogens: *Trichothecium roseum*, *Fusarium avenaceum,* and *Alternaria alternata* [86]. Polonio, et al. [87] observed that *D. citri* exhibited antifungal activity against *F. solani* and *Didymella bryoniae,* as well as antibacterial activity against *Staphylococcus aureus*. In addition, *Diaporthe* species isolated from the ornamental plant *Pachystachys lutea* were able to inhibit the growth of *F. oxysporum* and *Colletotrichum* sp. [88]. In a recent study, Verma, et al. [89] reported that *D. melonis* and *D. longicolla* suppressed the growth of *Corynespora cassiicola* and *F. solani* in dual-culture competition assays. *Phomopsis oblonga* (current name *P. valeta*) serves as a natural biocontrol agent for Dutch elm disease by acting as a feeding deterrent for elm bark beetles [90]. Our previous studies have shown that *D. maritima* isolated from *Picea* produces antiinsectant and antifungal metabolites [49]. Recently, *D. miriciae*, an endophytic fungus isolated from tropical medicinal plants, has been shown to produce cytochalasins, which have antifungal and antagonistic activity against plant pathogens [82].

### 4.3. Endophyte Diversity and Ecology

The twelve fungal endophyte strains isolated in this study represent seven genera in six families and five orders in the phylum Ascomycota. Six of the twelve strains are classified as *Diaporthe*, a speciose genus consisting of endophytes, saprotrophs, and pathogens that colonize a wide range of host plants [35]. This genus has more than 800 described species, and more than 950 species in its asexual state (formerly *Phomopsis*) [91]. Two of the isolated strains (En20-4 and En01-1) appear to represent putative novel species, based on their phylogenetic distinction from described species with sequences available in GenBank. Four strains are closely related to *D. vacuae*, a recently described species in the *D. eres* complex that has been associated with dieback and twig blight of *Vaccinium corymbosum* [54]. These strains also had similar extrolite profiles (Appendix A). Recent work resulted in *D. vacuae* and other related species in the *D. eres* complex being synonymized with *D. eres* [92]. En20-4. species are commonly associated with diseases of grapevine worldwide, notably Phomopsis cane and leaf spot, which are attributed to *D. ampelina* (=*D. viticola*), and symptoms such as stem and branch dieback, perennial cankers, vascular discoloration, and rachis necrosis [93,94,95,96]. Phomopsis cane and leaf spot can affect most parts of the grapevine, including the flowers and berries, and its potential to cause large crop losses often leads to the application of fungicides and other control measures [97]. Although *D. eres* causes minor disease in a broad range of hosts, it can also result in serious canker disease in grapevines, apples, and blueberries [54]. For example, strains of *D. eres* isolated from symptomatic apple rootstocks in the same region as this study (Annapolis Valley, NS) caused necrosis, cankers, and eventually death in young apple plants within four weeks [24]. Other *Diaporthe* species show varying degrees of virulence in grapevines; for example, the closely related *D. ampelina*, *D. hispaniae*, and *D. hungariae* are highly virulent, while other species, such as *D. bohemiae*, appear to be avirulent endophytes. Overall, most *Diaporthe* species, including *D. eres*, tested in grapevine pathogenicity experiments show some degree of virulence [93,98,99]. Reveglia, et al. [100] demonstrated that a *D. eres* strain isolated from symptomatic grapevine wood produced phytotoxic secondary metabolites; one phytotoxin, nectriapyrone, was identified in several of our *Diaporthe* strains. Guarnaccia, Groenewald, Woodhall, Armengol, Cinelli, Eichmeier, Ezra, Fontaine, Gramaje and Gutierrez-Aguirregabiria [39] reported that *D. eres* was the *Diaporthe* species most commonly isolated in grapevines sampled in eight countries. Taken together, the fact that *Diaporthe* represents the majority of endophytes isolated in this study is not unexpected. Its presence does not necessarily reflect the overall health of the sampled grapevines, given its frequency of occurrence on asymptomatic grapevines and potential host resistance [101]. However, it is notable that Kernaghan, Mayerhofer and Griffin [20] did not report any *Diaporthe* endophytes in wild and hybrid *Vitis* leaves sampled from wild grapes and vineyards in eastern Canada. Although pathogenicity experiments were not conducted in this study, the virulence of En20-4 and En01-1 should be assessed in the future, especially considering the identification of the phytotoxin nectriapyrone.

Although *Gnomoniopsis paraclavulata* (En61-1) was isolated from grapevine leaves, this species is best known from *Quercus* spp. in the US and Europe, where it is found in the stems, wood, asymptomatic leaves, overwintering leaves, leaf litter, and acorns [56,102,103,104]. The most dominant species isolated from asymptomatic and diseased stems of *Quercus robur* in Poland included *G. paraclavulata* (and, incidentally, *D. eres* and *C. fioriniae*) [62,105]. Subsequent pathogenicity tests on young *Q. robur* seedlings demonstrated that *G. paraclavulata* was the most pathogenic species tested, causing dieback symptoms and small lesions on stems. Costa, et al. [106] also identified *G. paraclavulata* in symptomatic and declining *Q. suber* in Portugal, while Tosi, et al. [107] implicated *G. paraclavulata* in chestnut bud and shoot blight (*Castanea sativa*). Given *G. paraclavulata*’s apparently narrow association with *Quercus,* or Fagaceae in general, its isolation from grape leaves is somewhat unexpected, and its interaction with this host is unknown.

*N. niveniae* (En61-2) is related *to Phaeomoniella chlamydospora* and is believed to be one of the primary causal agents of Petri disease and esca, two serious grapevine trunk diseases. Other Phaeomoniellales taxa have recently been described as a result of spore-trapping efforts in vineyards but do not appear to be pathogenic [33]. Similarly, *Neophaeomoniella* spp. has not been implicated in grapevine disease, although *N. niveniae* has been identified from symptomatic plants, albeit in low relative abundance [108]. Interestingly, *N*. *niveniae* was isolated from wild olive trees (*Olea europaea* subsp. *cuspidata*) in South Africa and was found to be present in 7.1% of the trees sampled; two strains exhibited low-to-zero virulence and intermediate virulence, respectively, when inoculated on shoots of ‘Frantoio’ olive trees (*O. europaea* L. subsp. *europaea*) in South Africa [67,109]. *N. niveniae* was first described from leaves of *Nivenia stokoei* in South Africa, where it was isolated from leaf spots, but the causality of this was not tested [66]. Our isolation of *N. niveniae* is likely the first record in Canada, and its interactions with grapevines are unknown. Interestingly, among the tested strains, En61-2 produced a number of dipeptides, all of which contain Ile or Leu. Although fungi are known to produce a wide range of cyclic dipeptides [110], all the ones reported here are linear and have unknown bioactivity.

Species of *Colletotrichum* have been implicated in various diseases of agricultural crops and other plants, such as bitter rot of pears and apples [111,112], anthracnose of apples [113], fruit rot of cranberries and blueberries [31], anthracnose fruit rot of raspberries [114], leaf spot disease of walnuts [115], anthracnose of eggplants [116], seedling blight of poison ivy (*Toxicodendron radicans)* [117], and leaf blight of *Mahonia aquifolium* [118]. Bitter rot of apples, for example, is considered a major disease of apples in warm and humid regions [104]. On the other hand, several species in this genus are capable of forming symbiotic and mutualistic relationships with plant species [119,120]. *C. fioriniae* was isolated as a leaf endophyte from apple trees in orchards and from various plant species growing nearby, including angiosperm trees, wild grapes, and others, leading the authors to hypothesize that the species’ main ecological role was indeed that of a leaf endophyte [104]. In the same study, the *C. fioriniae* strains isolated were also found to be pathogenic in apple fruit. Recently, *C. fioriniae* was reported as the causal agent of grapevine anthracnose in New York [121], and it is also associated with grapevine ripe rot [122]. *C. fioriniae* not only has been widely reported as an endophyte and pathogen in a broad range of hosts but is also cited as an entomopathogen of the elongate hemlock scale (*Fiorinia externa*), the origin of its specific epithet [123,124,125]. Therefore, *C. fioriniae* may act as a benign or potentially beneficial foliar endophyte in some plant hosts, while also reducing the quality of and accelerating decay in fruit. *C*. *gloeosponoides* has also been reported to cause epizootics of the scale insect *Orthezia praelonga*, a major pest of citrus in Brazil [124]. 

The family Xylariaceae contains a large number of commonly reported endophytes in a very broad range of host plants, including bryophytes, liverworts, angiosperms, and conifers, and even occurs in the form of endolichenic fungi [126]. These fungi are not only ubiquitous endophytes but also remarkably prolific producers of bioactive natural products [63,127]. The genus *Nemania* consists of approximately 80 named species (www.indexfungorum.org, accessed on 20 November 2023), including species characterized as endophytes producing described natural products [65,128,129,130,131]. *N. serpens* and *Nemania* spp. have occasionally been isolated as endophytes of cultivated and wild grapes [128,132]. We isolated *N. aureolutea* (En25-1), a rarely observed species that is closely related to the common *N. aenea*. Our strain produced the characteristic slow-growing orange colonies on PDA, as described by Fournier, Lechat and Ribes Ripoll [61] (Figure 3A). Little is known about *N. aureolutea* because it is rarely collected, with a small number of reports from Europe, the continental US, and Hawaii, where it grows in the wood, and sometimes on the outer bark, of *Acer*, *Corylus*, *Populus*, *Quercus*, and *Salix*; most collections appear to be from *Quercus* [61,133]. Our strain may be the first report from Canada; however, *N. aureolutea* is likely more common than perceived, and some specimens identified as *N. aenea* are actually *N. aureolutea* [134].

The genus *Sphaerulina* comprises more than 200 named species (www.indexfungorum.com, accessed on 20 November 2023), with some causing leaf spots on various plant species; notably, *S. musiva* is responsible for one of the most damaging diseases of poplar in northeastern and north-central North America [135]. *S. amelanchier* (En26-1) was originally described from *Amelanchier* sp. leaf litter in the Netherlands and has also been isolated from leaves of unidentified species of *Betula*, *Castanea*, and *Quercus* [72,136]. Given the infrequent reports, little is known about the ecology of *S. amelanchier*. A related species, *S. vaccinii*, causes a common leaf spot and stem canker disease in lowbush blueberry (*Vaccinium angustifolium*) in eastern Canada and the US [137,138,139]. We also isolated another Mycosphaerellaceae species, a strain (En60) of an unknown *Ramularia* species that is a sister species to *R. heraclei* and *R. lamii* var. *lamii*. *Ramularia* are also well-known and common associates of leaf spot diseases of varying significance in a broad range of plant hosts. For example, Ramularia leaf spots of sugar beet and barley, caused by *R. beticola* and *R. collo-cygni*, respectively, are serious emerging disease [140,141]. Species of *Ramularia* have been reported as endophytes of grapevines [20], while a *Ramularia* sp. isolated from *Rumex gmelini* Turcz has been noted to produce the bioactive secondary metabolite chrysophanol [142]. According to the results of the phylogenetic analysis and the lack of similarity to available sequences, *Ramularia* sp. (En60) is possibly a novel species.

In general, the host preferences of the endophyte species that we isolated from grapevines are either demonstrably broad (e.g., *Colletotrichum fioriniae*, *D. eres*), suggestive of a broader host range than currently known (e.g., *G. paraclavulata*, *Nemania aureolata, Neophaeomoniella niveniae*), or are unknown due to possible novelty (e.g., *Diaporthe* spp. En20-4 and En01-1, *Ramularia* sp.). Many of these endophytes produce slimy, bright yellow-orange masses of conidia that are presumably spread via rain splash and water runoff and potentially vectored by insects, including pollinators [33,143,144]. Endophytes in the Xylariaceae family (e.g., *Nemania aureolata*) may be transmitted horizontally to new leaves via dry, airborne conidia and ascospores. While grapevines are probably a secondary host for *N. aureolata*, endophytes occurring on leaves as foliar endophytes may colonize primary hosts (i.e., host substrate that supports sexual reproduction, such as hardwood trees) through direct contact (viaphytism) [145]. Therefore, the senescent and overwintered leaves and wood of both grapevines and other distantly related plant hosts may serve as inoculum for new infections. Vertical transmission of endophytes is also possible in some cases; for example, as demonstrated or suggested in *Diaporthe*, *Ramularia*, and *G. paraclavulata* [103,146,147].

Some grapevine endophytes are potentially opportunistic pathogens, such as *Diaporthe* spp. and *C. fioriniae*, which may affect vines, leaves, and/or fruit. Other species may be commensal or mutualistic, such as *N. aureolata* or even *C. fioriniae*, an endophyte that may protect against insect herbivory and, as demonstrated in other *Colletotrichum* species, provide protection from plant pathogens [31,148,149,150]. Interactions between endophytes and their hosts span the mutualist–pathogen continuum and depend on the host’s status at the given moment [151]. Consequently, the overall effect of endophyte colonization on host fitness may be difficult to generalize; for example, *C. fioriniae* could benefit its host by protecting leaves against insect herbivory, but its colonization of fruit could reduce seed dispersal by altering signals that attract animal seed dispersers, i.e., by reducing palatability and altering nutrient content [152]. Conversely, the fungal infection of fruit may actually increase dispersal; for example, by increasing levels of attractive volatiles, in which case an anthropocentric view of fruit diseases may lead to the misinterpretation of a fungus’ effects on its host’s fitness in natural systems [153].

In conclusion, this work was successful in identifying a number of promising endophytes, as well as the metabolites they produce, in grapevines with a mix of European and North American ancestry growing ferally in a challenging environment. However, this work is only an initial step towards the ultimate goal: the discovery of novel growth-promoting and biocontrol agents. While a number of the isolates found in this study showed promise in their suppression of *Botrytis* using ex situ competition assays, the development of molecular markers followed by in situ trials is needed. In future work, inoculation trials should be performed on healthy grapevines to ensure prospective endophytes are not pathogenic and to assess their growth promotion potential relative to control plants. Finally, once confirmed as an endophyte, control and inoculated host plants should be challenged with *Botrytis* and other common grapevine pathogens to assess their in planta biocontrol capabilities in a controlled environment followed by field trials. Several endophytes have been shown to alter plant hormones or metabolites, and their effect on the pathogen and disease suppression in situ could be very different from the effect of endophytes on pathogen suppression in ex situ competition assays. Also, the inoculated endophyte will have an effect on the microbiome of the plants, which could indirectly affect pathogen and disease suppression, so inoculating the plants with these endophytes in the field trials will help to fully understand this interaction. Future work may include expanding sampling efforts of both the host and isolated taxa while also including fungi from the phylloplane.

## 5. Significance Statement

Leaf samples collected from over 80 feral grapevine sites were used to isolate a range of fungal endophytes. Endophyte identification, competition assays against a known pathogen, and metabolite screening were used as an initial step towards understanding how these microbes may be utilized as biocontrol agents in commercial plantings.

## Figures and Tables

**Figure 1 jof-10-00105-f001:**
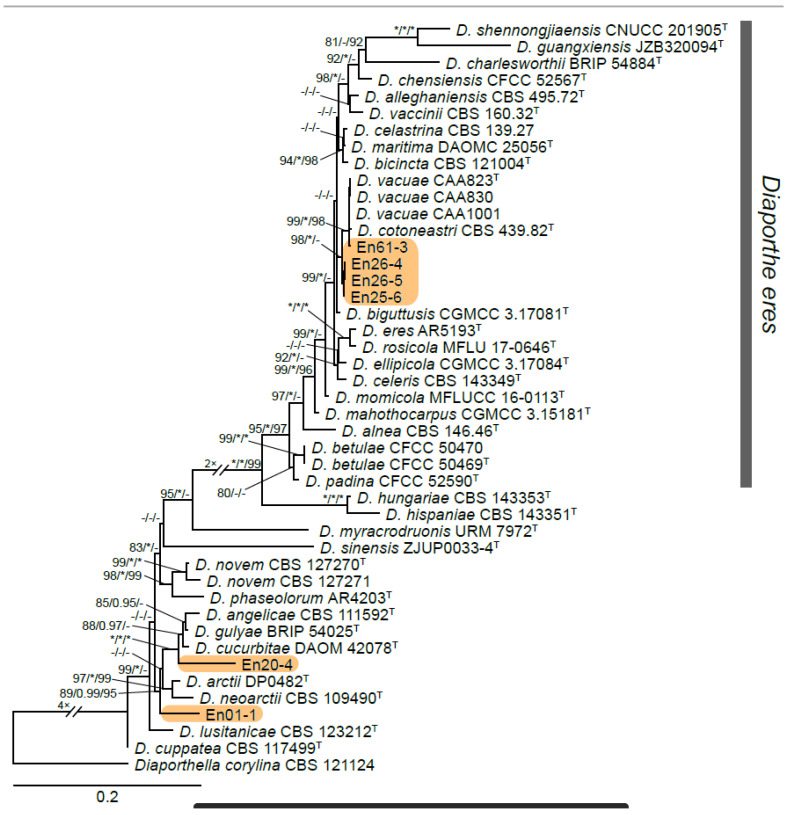
IQ-TREE maximum likelihood consensus tree inferred from the combined TUB2, TEF1-α, and ITS sequence alignment for *Diaporthe*. Strains representing species formally in the *D. eres* species complex are now synonymized as *D. eres*. Strains originating from this study are highlighted by an orange rectangular box. Support values at the nodes correspond to SH-aLRT (≥80%), aBayes (≥0.95), and ultrafast bootstrap (≥95%) support values; an asterisk (*) indicates full support (100% or 1.0), and a hyphen (-) indicates support lower than the significant values listed for each branch test. The tree is rooted to *Diaporthella corylina* (CBS 121124). The scale bar shows the expected number of nucleotide substitutions per site. Ex-type and ex-epitype strains are indicated by the symbol ^T^.

**Figure 2 jof-10-00105-f002:**
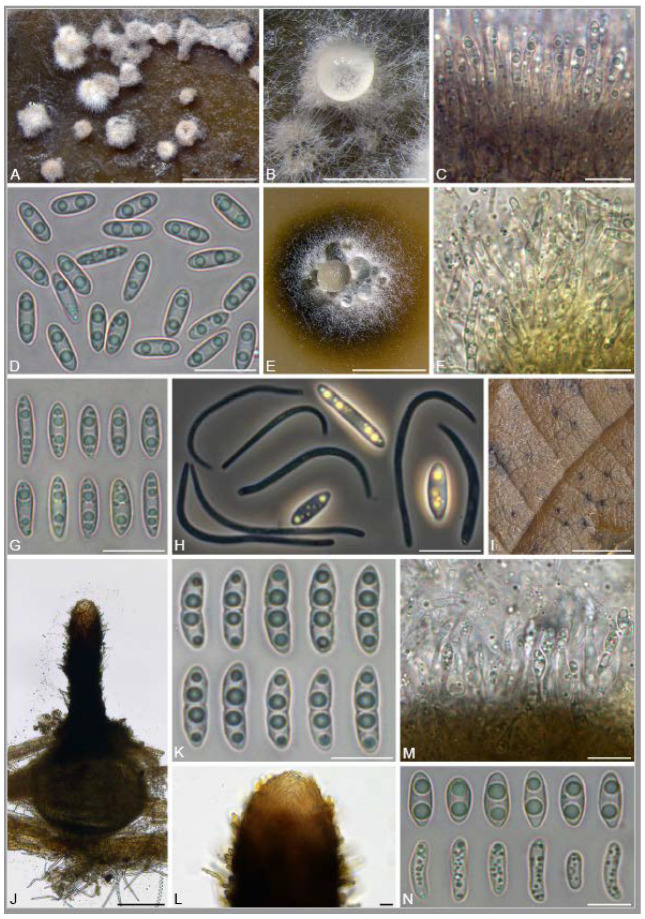
Morphology of *Diaporthe* spp. (**A**–**D**) *D. eres* (En25-6) on WA with *Ilex aquifolium* leaf. (**A**) Conidiomata; (**B**) conidioma with exuded droplet containing alpha conidia (on *I. aquifolium* leaf); (**C**) conidiogenous cells and alpha conidia; (**D**) alpha conidia; (**E**–**H**) *D.* aff. *gulyae* (En20-4); (**E**) conidiomata with conidial droplets; (**F**) conidiogenous cells; (**G**) alpha conidia; (**H**) alpha and beta conidia; (**I**–**N**) *Diaporthe* sp. (En01-1) on WA with *Rubus armeniacus* leaf; (**I**) perithecia on leaf surface; (**J**) whole perithecium; (**K**) ascospores; (**L**) apex of perithecial neck; (**M**) alpha conidia (top row) and gamma conidia (bottom row); (**N**) conidiogenous cells. Scale bars: (**A**,**E**,**I**) **=** 1 mm; (**B**) = 500 μm; (**C**,**D**,**F**–**H**,**K**–**N**) = 10 μm; (**J**) = 100 μm.

**Figure 3 jof-10-00105-f003:**
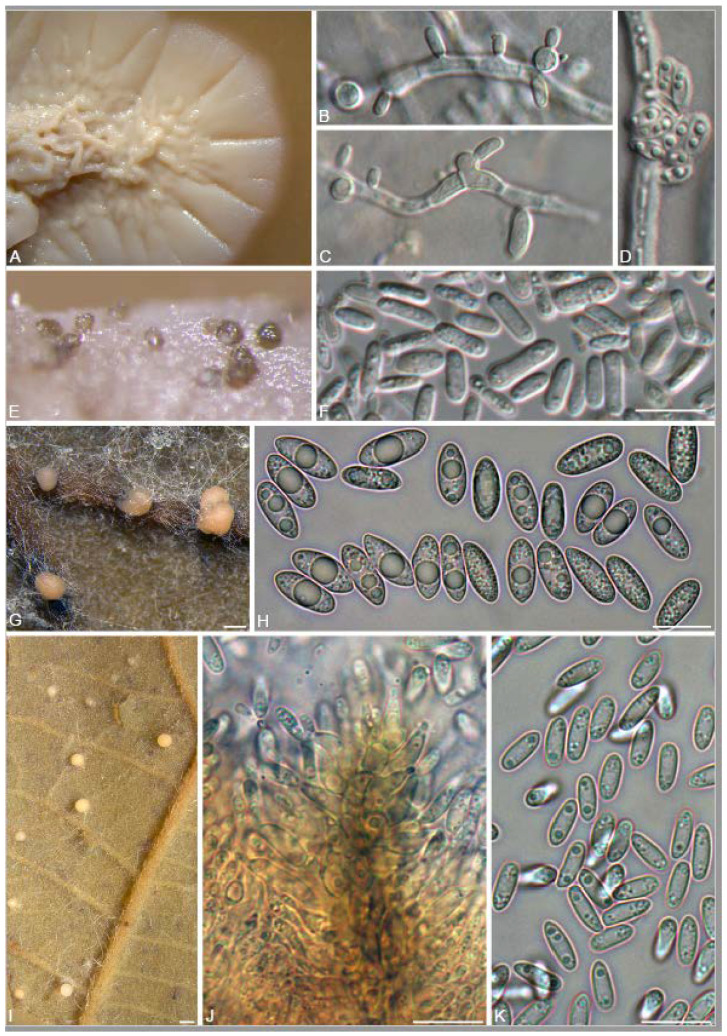
Morphology of *Neophaeomoniella niveniae, Colletotrichum fioriniae,* and *Gnomoniopsis paraclavulata*. (**A**–**F**) *N. niveniae* (En61-2); (**A**) two-week-old colony on V8A. (**B**,**C**) intercalary conidiogenous cells and conidia on oatmeal agar (OA); (**D**) slimy mass of conidia on hypha (OA); (**E**) conidiomata (OA); (**F**) conidia (OA); (**G**,**H**) *C. fioriniae* (En25-3) on water agar with *Rubus armeniacus* leaf; (**G**) conidiomata with salmon-colored masses of conidia on leaf veins; (**H**) conidia; (**I**–**K**) *G. paraclavulata* (En61-1) on WA with *Rubus armeniacus* leaf; (**I**) conidiomata with orange masses of conidia; (**J**) conidiogenous cells and conidia; (**K**) conidia. Scale bars: (**A**) = 1 mm; (**B**–**D**,**F**,**H**,**J**,**K**) = 10 μm; (**E**) = 100 μm. (**G**,**I**) = 1 mm.

**Figure 4 jof-10-00105-f004:**
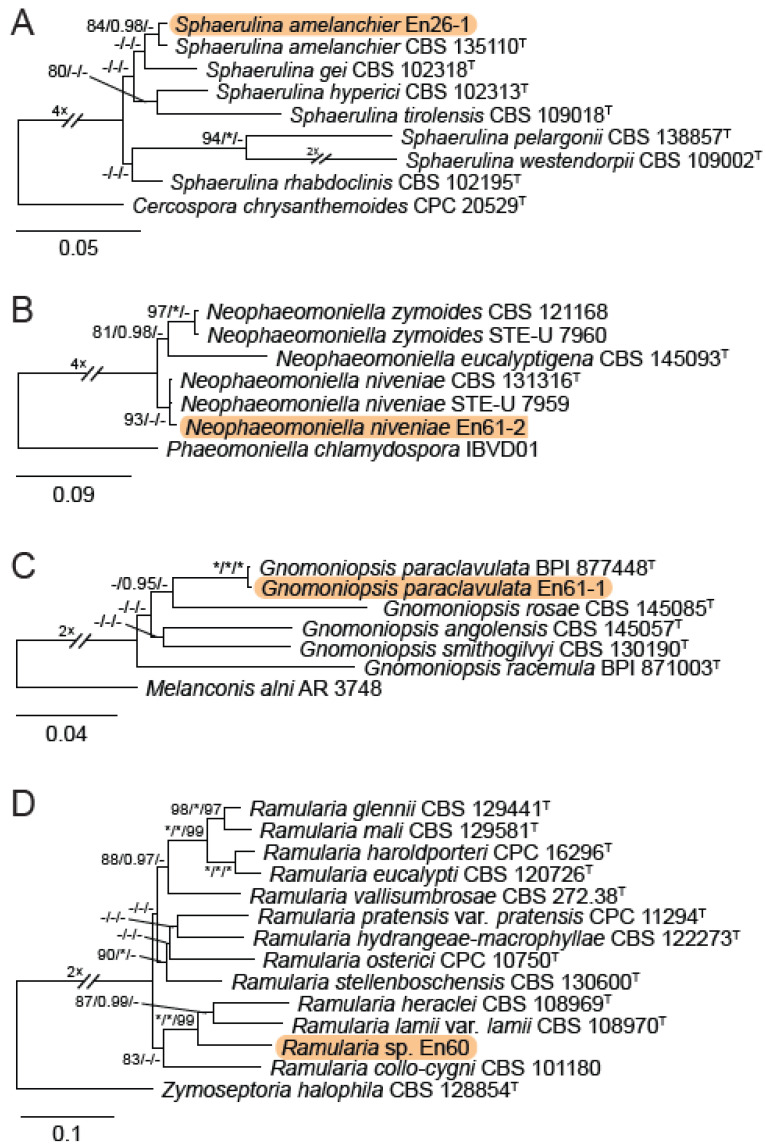
IQ-TREE maximum-likelihood consensus trees, with the strains identified in the study highlighted by orange rectangular boxes. (**A**) IQ-TREE maximum likelihood consensus tree inferred from the TEF1-α sequence alignment for *Sphaerulina,* rooted to *Cercospora chrysanthemoides* (CPC 20529), and strain En26-1 originating from this study is highlighted by an orange rectangular box. The tree is rooted to *Cercospora chrysanthemoides* (CPC 20529). (**B**) IQ-TREE maximum likelihood consensus tree inferred from the combined ITS and TEF1-α sequence alignment for *Neophaeomoniella*. The strain (En61-2) originating from this study is highlighted by an orange rectangular box. The tree is rooted to *Phaeomoniella chlamydospora* (IBVD01). (**C**) IQ-TREE maximum likelihood consensus tree inferred from the combined ITS and RPB2 sequence alignment for *Gnomoniopsis*. The strain (En61-1) originating from this study is highlighted by an orange rectangular box. The tree is rooted to *Melanconis alni* (AR 3748). (**D**) IQ-TREE maximum likelihood consensus tree inferred from the combined ACT, ITS, and RPB2 sequence alignment for *Ramularia*. The strain (En60) originating from this study is highlighted by an orange rectangular box. The tree is rooted to *Zymoseptoria halophila* (CBS 128854). Support values at the nodes correspond to SH-aLRT (≥80%), aBayes (≥0.95), and ultrafast bootstrap (≥95%) support values; an asterisk (*) indicates full support (100% or 1.0) and a hyphen (-) indicates support lower than the significant values listed for each branch test. The scale bar shows the expected number of nucleotide substitutions per site. Ex-type and ex-epitype strains are indicated by the symbol ^T^.

**Figure 5 jof-10-00105-f005:**
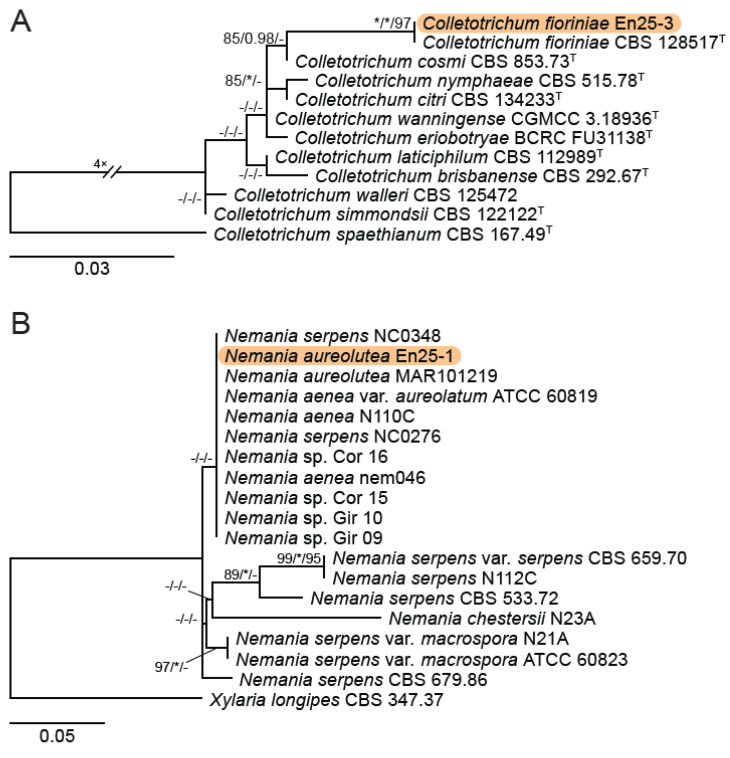
IQ-TREE maximum likelihood consensus trees (**A**) IQ-TREE maximum likelihood consensus tree inferred from the CHS-1 sequence alignment for *Colletotrichum.* The strain (En25-3) originating from this study is highlighted by an orange rectangular box. The tree is rooted to *Colletotrichum spaethianum* (CBS 167.49). (**B**) IQ-TREE maximum likelihood consensus tree inferred from the ITS sequence alignment for *Nemania*. The strain (En25-1) originating from this study is highlighted by an orange rectangular box. The tree is rooted to *Xylaria longipes* (CBS 347.37). Support values at the nodes correspond to SH-aLRT (≥80%), aBayes (≥0.95), and ultrafast bootstrap (≥95%) support values; an asterisk (*) indicates full support (100% or 1.0), and a hyphen (-) indicates support lower than the significant values listed for each branch test. The scale bar shows the expected number of nucleotide substitutions per site. Ex-type and ex-epitype strains are indicated by the symbol ^T^.

**Figure 6 jof-10-00105-f006:**
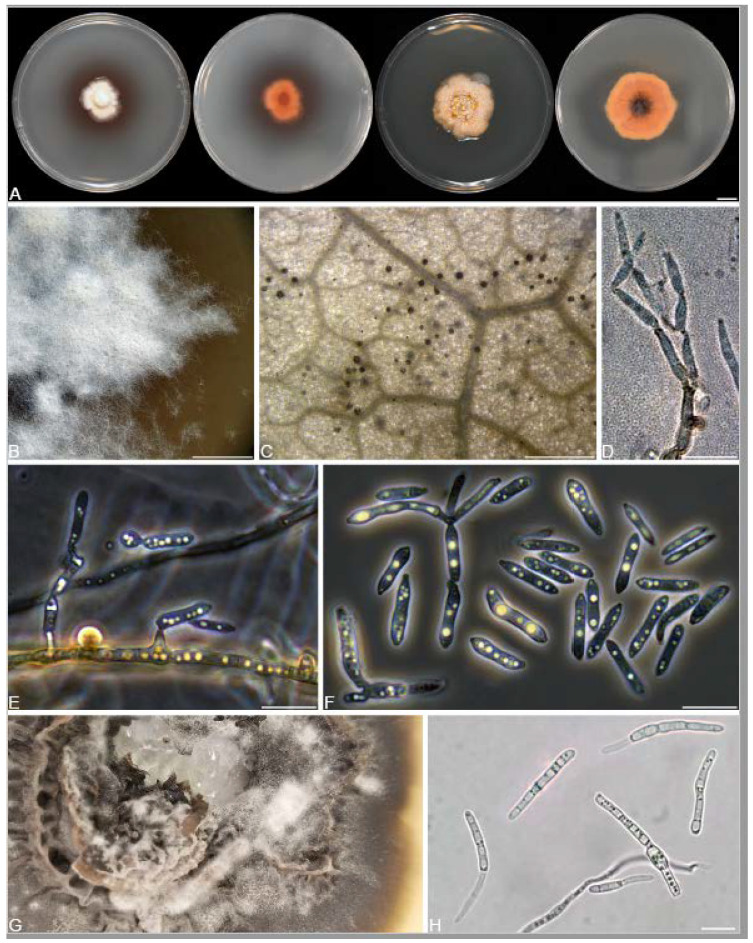
Morphology of *Nemania aureolutea*, *Ramularia* sp., and *Sphaerulina amelanchier*. (**A**) *Nemania aureolutea* (En25-1) on PDA in the first two plates show the front and reverse of two-week-old colonies and, the last two plates show the front and reverse of four-week-old colonies. (**B**–**F**) *Ramularia* sp. (En60); (**B**) two-week-old colony on V8A; (**C**) conidiomata on WA with *Ilex aquifolium* leaf; (**D**–**F**) conidiophores, conidiogenous cells, and conidia; (**G**) *Sphaerulina amelanchier* En26-1 on MEA; (**H**) conidia. Scale bars: (**A**,**B**) **=** 1 cm; (**C**) = 500 μm; (**D**–**F**,**H**) = 10 μm.

**Figure 7 jof-10-00105-f007:**
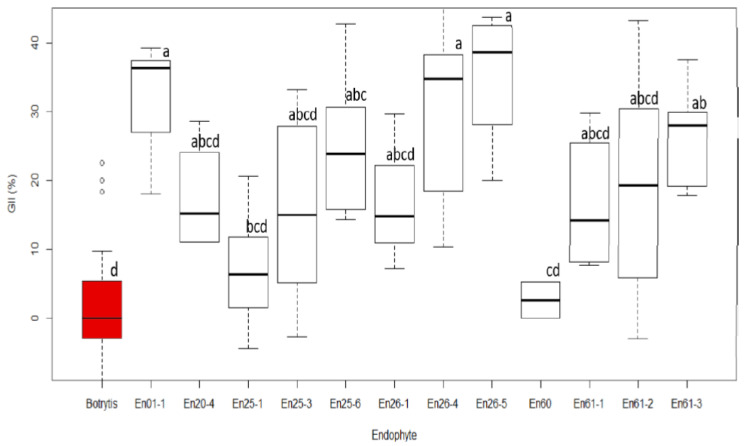
A box and whisker plot showing the percent growth inhibition index (GII) of *Botrytis cinerea* by itself (red), which acted as a control, and by different grape fungal endophytes isolated from ancient wine grapes. The percent GII was calculated as (HP-VP)/HP × 100, where HP is the mean of horizontal growth (mean radius) of the pathogen and VP is the vertical growth of the pathogen (mean radius) towards the endophyte. Values with shared letter groupings are not significantly different, as measured via a post hoc Tukey multiple means comparison.

**Table 1 jof-10-00105-t001:** GenBank accession phylogenesis.

Species	Isolate	Host	Location	GenBank Accession No.							Reference
				TUB2	ITS	TEF1-α	RPB2	ACT	CHS-1	LSU	
*Cercospora chrysanthemoides*	CPC 20529	*Chrysanthemoides monilifera*	South Africa	—	—	KC005813	—	—	—	—	[30]
*Colletotrichum brisbanense*	CBS 292.67	*Capsicum annuum*	Australia	—	—	—	—	—	JQ948952	—	[31]
*Colletotrichum citri*	CBS 134233	*Citrus aurantifolia*	China	—	—	—	—	—	KY856138	—	[32]
*Colletotrichum cosmi*	CBS 853.73	*Cosmos sp.*	Netherlands	—	—	—	—	—	JQ948935	—	[31]
*Colletotrichum eriobotryae*	BCRC FU31138	*Eriobotrya japonica*	Taiwan	—	—	—	—	—	MN191653	—	[33]
*Colletotrichum fioriniae*	En25-3	*Vitis sp. ‘Clinton’*	Canada	—	MZ127182	—	OK431476	—		OK380951	This study.
*Colletotrichum fioriniae*	CBS 128517	*Fiorinia externa*	USA	—	—	—	—	—	JQ948953	—	[31]
*Colletotrichum laticiphilum*	CBS 112989	*Hevea brasiliensis*	India	—	—	—	—	—	JQ948950	—	[31]
*Colletotrichum nymphaeae*	CBS 515.78	*Nymphaea alba*	Netherlands	—	—	—	—	—	JQ948858	—	[31]
*Colletotrichum simmondsii*	CBS 122122	*Carica papaya*	Australia	—	—	—	—	—	JQ948937	—	[31]
*Colletotrichum spaethianum*	CBS 167.49	*Hosta sieboldiana*	Germany	—	—	—	—	—	GU228297	—	[34]
*Colletotrichum walleri*	CBS 125472	*Coffea* sp.	Vietnam	—	—	—	—	—	JQ948936	—	[31]
*Colletotrichum wanningense*	CGMCC 3.18936	*Hevea brasiliensis*	China	—	—	—	—	—	MZ352012	—	Liu et al. (Direct Submission)
*Diaporthe* aff. *gulyae*	En20-4	*Vitis sp. unknown*	Canada	OK383388	MZ127185		—	—	—	OK380954	This study
*Diaporthe alleghaniensis*	CBS 495.72	*Betula alleghaniensis*	Canada	KC843228	NR_103696	KC343733	—	—	—	—	[35]
*Diaporthe alnea*	CBS 146.46	*Alnus* sp.	Unknown	KC343976	NR_147525	KC343734	—	—	—	—	[35]
*Diaporthe angelicae*	CBS 111592	*Heracleum sphondylium*	Austria	KC343995	KC343027	KC343753	—	—	—	—	[35]
*Diaporthe arctii*	DP0482	*Arctium lappa*	Austria	KJ610891	KJ590736	KJ590776	—	—	—	—	[36]
*Diaporthe betulae*	CFCC 50470	*Betula platyphylla*	China	KT733021	KT732951	KT733017	—	—	—	—	[37])
*Diaporthe betulae*	CFCC 50469	*Betula platyphylla*	China	KT733020	NR_147578	KT733016	—	—	—	—	[37]
*Diaporthe bicincta*	CBS 121004	*Juglans* sp.	USA	KC344102	NR_147526	KC343860	—	—	—	—	[35]
*Diaporthe biguttusis*	CGMCC 3.17081	*Lithocarpus glaber*	China	KF576306	NR_147533	KF576257	—	—	—	—	[38]
*Diaporthe celastrina*	CBS 139.27	*Celastrus scandens*	Unknown	KC344015	NR_152457	KC343773	—	—	—	—	[35]
*Diaporthe celeris*	CBS 143349	*Vitis vinifera*	UK	MG281190	MG281017	MG281538	—	—	—	—	[39]
*Diaporthe charlesworthii*	BRIP 54884m	*Rapistrum rugosum*	Australia	KJ197268	NR_147538	KJ197250	—	—	—	—	[40]
*Diaporthe chensiensis*	CFCC 52567	*Abies chensiensis*	China	MH121584	NR_165876	MH121544	—	—	—	—	[41]
*Diaporthe cotoneastri*	CBS 439.82	*Cotoneaster* sp.	UK	JX275437	MH861511	GQ250341	—	—	—	—	[42,43,44]
*Diaporthe cucurbitae*	DAOM 42078	*Cucumis* sp.	Canada	KP118848	KM453210	KM453211	—	—	—	—	[36]
*Diaporthe cuppatea*	CBS 117499	*Aspalathus linearis*	South Africa	KC344025	MH863021	KC343783	—	—	—	—	[35,42]
*Diaporthe ellipicola*	CGMCC 3.17084	*Lithocarpus glaber*	China	KF576294	NR_147531	KF576245	—	—	—	—	[38]
*Diaporthe eres*	En61-3	*Vitis sp. ‘Isabella’*	Canada	OK383392	MZ127189		—	—	—	OK380958	This study
*Diaporthe eres*	En26-4	*Vitis sp. ‘Clinton’*	Canada	OK383390	MZ127187		—	—	—	OK380956	This study
*Diaporthe eres*	En26-5	*Vitis sp. ‘Clinton’*	Canada	OK383391	MZ127188		—	—	—	OK380957	This study
*Diaporthe eres*	En25-6	*Vitis sp. ‘Clinton’*	Canada	OK383389	MZ127186		—	—	—	MZ127186	This study
*Diaporthe eres*	AR5193	*Ulmus* sp.	Germany	KJ420799	KJ210529	KJ210550	—	—	—	—	[45]
*Diaporthe guangxiensis*	JZB320094	*Vitis vinifera*	China	MK500168	MK335772	MK523566	—	—	—	—	[46]
*Diaporthe gulyae*	BRIP 54025	*Helianthus annuus*	Australia	KJ197271	NR_111615	JN645803	—	—	—	—	[40,47,48]
*Diaporthe hispaniae*	CBS 143351	*Vitis vinifera*	Spain	MG281296	MG281123	MG281644	—	—	—	—	[39]
*Diaporthe hungariae*	CBS 143353	*Vitis vinifera*	Hungary	MG281299	MG281126	MG281647	—	—	—	—	[39]
*Diaporthe lusitanicae*	CBS 123212	*Foeniculum vulgare*	Portugal	KC344104	MH863279	KC343862	—	—	—	—	[35,42],
*Diaporthe mahothocarpus*	CGMCC 3.15181	*Lithocarpus glabra*	China	KF576312	KC153096	KC153087	—	—	—	—	
*Diaporthe maritima*	DAOMC 25056	*Picea rubens*	Canada	KU574615	NR_152463	KU552023	—	—	—	—	[49]
*Diaporthe momicola*	MFLUCC 16-0113	*Prunus persica*	China	KU557587	NR_172386	KU557631	—	—	—	—	[50]
*Diaporthe myracrodruonis*	URM 7972	*Astronium urundeuva*	Brazil	MK205291	NR_163320	MK213408	—	—	—	—	[15]
*Diaporthe neoarctii*	CBS 109490	*Ambrosia trifida*	USA	KC344113	NR_111854	KC343871	—	—	—	—	[35,47]
*Diaporthe novem*	CBS 127270	*Glycine max*	Croatia	KC344124	MH864503	KC343882	—	—	—	—	[35,42]
*Diaporthe novem*	CBS 127271	*Glycine max*	Croatia	KC344125	MH864504	KC343883	—	—	—	—	[35,42]
*Diaporthe padina*	CFCC 52590	*Prunus padus*	China	MH121604	NR_165879	MH121567	—	—	—	—	[41]
*Diaporthe phaseolorum*	AR4203	*Phaseolus vulgaris*	USA	KJ610893	KJ590738	KJ590739	—	—	—	—	[36]
*Diaporthe rosicola*	MFLU 17-0646	*Rosa* sp.	UK	MG843877	NR_157515	MG829270	—	—	—	—	[51]
*Diaporthe shennongjiaensis*	CNUCC 201905	*Juglans regia*	China	MN227012	MN216229	MN224672	—	—	—	—	[52]
*Diaporthe sinensis*	ZJUP0033-4	*Amaranthus* sp.	China	MK660447	MK637451	MK660449	—	—	—	—	[53]
*Diaporthe* sp.	En01-1	*Vitis riparia*	Canada	OK383387	MZ127184		—	—	—	OK380953	This study
*Diaporthe vaccinii*	CBS 160.32	*Oxycoccus macrocarpos*	USA	KC344196	NR_103701	KC343954	—	—	—	—	[35]
*Diaporthe vacuae*	CAA829	*Vaccinium corymbosum*	Portugal	MK837928	MK792306	MK828077	—	—	—	—	[54]
*Diaporthe vacuae*	CAA830	*Vaccinium corymbosum*	Portugal	MK837931	MK792309	MK828080	—	—	—	—	[54])
*Diaporthe vacuae*	CAA1001	*Quercus suber*	Portugal	MT309458	MT237172	MT309432	—	—	—	—	[54]
*Diaporthella corylina*	CBS 121124	*Corylus* sp.	China	KC343972	KC343004	KC343730	—	—	—	—	[35]
*Gnomoniopsis angolensis*	CBS 145057	Unknown plant sp.	Angola	—	MK047428	—	MK047539	—	—	—	[55]
*Gnomoniopsis paraclavulata*	BPI 877448	*Carpinus caroliniana*	USA	—	EU254839	—	EU219248	—	—		[56]
*Gnomoniopsis paraclavulata*	En61-1	*Vitis sp. ‘Isabella’*	Canada	OK383385	MZ127179	—	OK431474	—	—	OK380949	This study.
*Gnomoniopsis racemula*	BPI 871003	*Epilobium angustifolium*	USA	—	EU254841	—	EU219241	—	—	—	[56]
*Gnomoniopsis rosae*	CBS 145085	*Rosa* sp.	New Zealand	—	NR_161142	—	MK047547	—	—	—	[55]
*Gnomoniopsis smithogilvyi*	CBS 130190	*Castanea* sp.	Australia	—	MH865607	—	JQ910648	—	—	—	[30,42]
*Melanconis alni*	AR 3748	*Alnus viridis*	Austria	—	EU199195	—	EU199153	—	—	—	[57]
*Nemania aenea*	N110C	Unknown	Unknown	—	AJ390427	—	—	—	—	—	[58]
*Nemania aenea*	nem046	*Centaurea stoebe*	USA	—	EF589887	—	—	—	—	—	[59]
*Nemania aenea var. aureolatum*	ATCC 60819	*Quercus* sp.	Switzerland	—	AF201704	—	—	—	—	—	[60]
*Nemania aureolutea*	En25-1	*Vitis sp. ‘Clinton’*	Canada	OK383386	MZ127183	—	OK431477	OK431472	—	OK380952	This study
*Nemania aureolutea*	MAR101219	*Quercus canariensis*	Spain	—	MW136058	—	—	—	—	—	[61]
*Nemania chestersii*	N23A	Unknown	Unknown	—	AJ390430	—	—	—	—	—	[58]
*Nemania serpens*	NC0348	*Lecanora oreinoides*	USA	—	JQ761380	—	—	—	—	—	[62]
*Nemania serpens*	NC0276	*Diploschistes rampoddensis*	USA	—	JQ761314	—	—	—	—	—	[62]
*Nemania serpens*	N112C	Unknown	Unknown	—	AJ390436	—	—	—	—	—	[58]
*Nemania serpens*	CBS 533.72	*Corylus avellana*	Netherlands	—	FN428829	—	—	—	—	—	[63]
*Nemania serpens*	CBS 679.86	*Betula* sp.	Switzerland	—	KU683765	—	—	—	—	—	[62]
*Nemania serpens* var. *macrospora*	N21A	Unknown	Unknown	—	AJ390433	—	—	—	—	—	[58]
*Nemania serpens* var. *macrospora*	ATCC 60823	Unknown	Unknown	—	AF201707	—	—	—	—	—	[60]
*Nemania serpens* var. *serpens*	CBS 659.70	Soil	Canada	—	MH859890	—	—	—	—	—	[42]
*Nemania* sp.	Cor 16	*Nephroma laevigatum*	France	—	MG916993	—	—	—	—	—	[64]
*Nemania* sp.	Cor 15	*Nephroma laevigatum*	France	—	MG916992	—	—	—	—	—	[64]
*Nemania* sp.	Gir 10	*Nephroma laevigatum*	France	—	MG917014	—	—	—	—	—	[64]
*Nemania* sp.	Gir 9	*Nephroma laevigatum*	France	—	MG917013	—	—	—	—	—	[64]
*Neophaeomoniella eucalyptigena*	CBS 145093	*Eucalyptus pilularis*	Australia	—	NR_161148	MK047569	—	—	—	—	[55]
*Neophaeomoniella niveniae*	CBS 131316	*Nivenia stokoei*	South Africa	—	JQ044435	MN861682	—	—	—	—	[65,66]
*Neophaeomoniella niveniae*	STE-U 7959	*Olea europaea* subsp. *cuspidata*	South Africa	—	MT791053	MT787396	—	—	—	—	[67]
*Neophaeomoniella niveniae*	En61-2	*Vitis sp. ‘Isabella’*	Canada	OK383384	MZ127178		—	—	—	OK380948	This study
*Neophaeomoniella zymoides*	CBS 121168	*Prunus salicina*	South Africa	—	GQ154600	MN861679	—	—	—	—	[65,68]
*Neophaeomoniella zymoides*	STE-U 7960	*Olea europaea* subsp. *cuspidata*	South Africa	—	MT791054	MT787397	—	—	—	—	[67]
*Phaeomoniella chlamydospora*	IBVD01	*Vitis vinifera*	Brazil	—	KP213118	KP213113	—	—	—	—	[69]
*Ramularia collo-cygni*	CBS 101180	*Hordeum vulgare*	Austria	—	NR_154944	—	KX288543	KX287666	—	—	[70]
*Ramularia eucalypti*	CBS 120726	*Corymbia grandifolia*	Italy	—	KJ504792	—	KJ504663	KF253635	—	—	[71,72]
*Ramularia glennii*	CBS 129441	*Homo sapiens*	Netherlands	—	MH865235	—	KJ504433	KJ504640	—	—	[42,71]
*Ramularia haroldporteri*	CPC 16296	Unidentified plant	South Africa	—	NR_154911	—	KJ504637	KJ504430	—	—	[71]
*Ramularia heraclei*	CBS 108969	*Heracleum sphondylium*	Netherlands	—	NR_154948	—	KX288578	KX287702	—	—	[70]
*Ramularia hydrangeae-macrophyllae*	CBS 122273	*Hydrangea macrophylla*	New Zealand	—	NR_145125	—	KX288592	KX287716	—	—	[70]
*Ramularia lamii* var. *lamii*	CBS 108970	*Lamium album*	Netherlands	—	NR_154949	—	KX288620	KX287744	—	—	[70]
*Ramularia mali*	CBS 129581	*Malus* sp.	Italy	—	NR_156582	—	KJ504649	MH876894	—	—	[42,71]
*Ramularia osterici*	CPC 10750	*Ostericum koreanum*	South Korea	—	NR_154950	—	KX288642	KX287765	—	—	[70]
*Ramularia pratensis var. pratensis*	CPC 11294	*Rumex crispus*	South Korea	—	EU019284.2	—	KT216537	KF903599	—	—	[73,74,75]
*Ramularia* sp.	En60-1	*Vitis sp. ‘Marechal foch’*	Canada	OK383393	MZ127180	—	OK431478	OK431473	—	OK380959	This study
*Ramularia stellenboschensis*	CBS 130600	*Protea* sp.	South Africa	—	NR_145101	—	KX288676	KX287798	—	—	[42,70]
*Ramularia vallisumbrosae*	CBS 272.38	*Narcissus* ‘Golden Spur’	UK	—	NR_154953	—	KX288698	KX288699	—	—	[70]
*Sphaerulina amelanchier*	En26-1	*Vitis sp. ‘Clinton’*	Canada	—	MZ127181		OK431475	—	—	OK380950	This study.
*Sphaerulina amelanchier*	CBS 135110	*Amelanchier* sp.	Netherlands	—	—	KF253543	—	—	—	—	[72]
*Sphaerulina gei*	CBS 102318	*Geum urbanum*	Netherlands	—	—	KF253560	—	—	—	—	[72]
*Sphaerulina hyperici*	CBS 102313	*Hypericum* sp.	Netherlands	—	—	KF253563	—	—	—	—	[72]
*Sphaerulina pelargonii*	CBS 138857	*Pelargonium* sp.	South Africa	—	—	KP004506	—	—	—	—	[72]
*Sphaerulina rhabdoclinis*	CBS 102195	*Pseudotsuga menziesii*	Germany	—	—	KF253578	—	—	—	—	[72]
*Sphaerulina tirolensis*	CBS 109018	*Rubus idaeus*	Austria	—	—	KF253585	—	—	—	—	[72]
*Sphaerulina westendorpii*	CBS 109002	*Rubus* sp.	Netherlands	—	—	KF253588	—	—	—	—	[72]
*Xylaria longipes*	CBS 347.37	Unknown	Unknown	—	MH855925	—	—	—	—	—	[42]
*Zymoseptoria halophila*	CBS 128854	*Hordeum vulgare*	Iran	—	MH865126	—	KX348110	KF253946	—	—	[30,42,72]

## Data Availability

Data are contained within the article and Appendix A.

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
