# Peer review of "Fungal Endophytes: Discovering What Lies within Some of Canada’s Oldest and Most Resilient Grapevines"

_jof, 2024, doi:10.3390/jof10020105_

Round 1

Reviewer 1 Report

Comments and Suggestions for Authors

Summary

This is largely a carefully executed study, with appropriate controls and careful relation to past literature. It will be a useful contribution to the literature.

Minor points

In view of recent studies it would be useful to state how the species of Botrytis was verified.

"ANOVA was later performed on this group to test for significance".  Significance of what?  What do the details of the boxes  in the figure show (median or mean etc).   How is the boxplot for Botrytis constructed?  Surely it always grows as much as itself?  We shouldn't have to guess all this.

General comments

The discussion is very long and seems poorly connected to the ambitious aims set out in lines 84-88.  The conclusions of the study do not appear to provide the required information to accomplish these aims, but you don’t actually say that.

It seemed to me that the results could be characterised as being inconclusive in relation to the aims, but being useful data to spark further studies.  If this is incorrect, would it be possible to be more explicit as to how the study moves understanding of the general area forward?

Much space is taken up with material from the prior literature;  I suggest that some of this could be omitted or abbreviated.

The slight reductions in growth of botrytis seem a weak start to finding new biocontrol agents, and perhaps it would have been helpful to compare with the range of reductions of botytis from phylloplane organisms.  How much competition did the botrytis in turn exert on the endophytes?

The study does bring up some general matters which might merit a brief mention in the introduction (or discussion). First, the microbiome can be rigorously defined so as to exclude the phylloplane, as here. But then, what if the effect of the microbiome on the host depends on phylloplane organisms?  Should sterilisation therefore be less stringent? Second, the method adopted here, largely excluding the host reactions in mediating an effect of the microbiome on disease processes, risks either missing a key interaction or creating an impossible multiplicity of combinations to test before the interactions are fully understood.  Third, to be sure what is going on, much more extensive sampling of both host and isolated organisms is needed - which may, of course, need more resource than is available. Comment on how to deal with this issue might be generally useful.

Author Response

Summary

This is largely a carefully executed study, with appropriate controls and careful relation to past literature. It will be a useful contribution to the literature.

Minor points

In view of recent studies it would be useful to state how the species of Botrytis was verified.

Response: This Botrytis is our lab strains. The strain used in this study has been verified by morphological and molecular characteristics as Botrytis cinerea. We have sequenced several genes such as ITS, β-tubulin, Cytb and SdhB gens from this Botrytis strain to confirm its identity and we have conducted pathogenicity on several plant species to make sure this strain is fully virulent.

"ANOVA was later performed on this group to test for significance".  Significance of what?  What do the details of the boxes  in the figure show (median or mean etc).   How is the boxplot for Botrytis constructed?  Surely it always grows as much as itself?  We shouldn't have to guess all this.

Response: Changes have been made to the manuscript in response to these excellent points. The competition assay Figure shows a “Box and whisker plot” (this detail now included). For this type of plot, the top and bottom lines of the box show the lower and upper quartiles, meaning that the box itself covers the interquartile interval (i.e., where 50% of the data is found). The horizontal line that splits the box in two is the median. The “whiskers” (horizontal lines attached by a vertical line to the box) show the lower and upper extremes of the data. The Botrytis data and plot were gathered and represented in the same way as the endophytes. Botrytis was added to a plate, instead of an endophyte, after the initial inoculation was already established. This process was replicated multiple times in the same way that the endophytes were, and the same measurements were taken. As a result of the numerous factors, including nuisance ones, that influence microbe growth on a plate, the growth of the second / new Botrytis colony was sometimes slightly drawn to, and other times slightly repelled by, the preexisting Botrytis colony (i.e., the “whiskers range from a GII of about -10 to 10). This gives the reader a sense of the intrinsic variability of the growth (i.e., or “noise”) in our tests. As the median was close to zero, this means the second Botrytis colony, i.e., our ‘control’, was, on average, not attracted or repelled by itself (which was a nice result, but was not necessarily a foregone conclusion).

General comments

The discussion is very long and seems poorly connected to the ambitious aims set out in lines 84-88.  The conclusions of the study do not appear to provide the required information to accomplish these aims, but you don’t actually say that.

Response: We agree with the reviewer that the discussion is a bit long but we think we have provided very useful information in the discussion about the isolated endophyte and also their potential role as pathogens in different plant species that will guide future studies in this direction.

It seemed to me that the results could be characterised as being inconclusive in relation to the aims, but being useful data to spark further studies.  If this is incorrect, would it be possible to be more explicit as to how the study moves understanding of the general area forward?

Response: We have added a concluding paragraph at the end of the Discussion section to show that data presented in this paper is very useful and will guide further studies.

Much space is taken up with material from the prior literature;  I suggest that some of this could be omitted or abbreviated.

The slight reductions in growth of botrytis seem a weak start to finding new biocontrol agents, and perhaps it would have been helpful to compare with the range of reductions of Botrytis from phylloplane organisms.  How much competition did the botrytis in turn exert on the endophytes?

Response: Thanks for raising this point, in our initial analysis of endophytes screening against botrytis, we occasionally observed some inhibitory effect of Botrytis on some endophytes isolates due to competition. We also measured the vertical growth of the endophyte (towards Botrytis) and horizontal growth of the endophytes towards edges of plates. In the 12 endophytes reported here we did not see any inhibitory effect of Botrytis on endophytes (Figure S1).  

The study does bring up some general matters which might merit a brief mention in the introduction (or discussion). First, the microbiome can be rigorously defined so as to exclude the phylloplane, as here. But then, what if the effect of the microbiome on the host depends on phylloplane organisms?  Should sterilisation therefore be less stringent? Second, the method adopted here, largely excluding the host reactions in mediating an effect of the microbiome on disease processes, risks either missing a key interaction or creating an impossible multiplicity of combinations to test before the interactions are fully understood.  Third, to be sure what is going on, much more extensive sampling of both host and isolated organisms is needed – which may, of course, need more resource than is available. Comment on how to deal with this issue might be generally useful.

Response: All these are good points. We have added a concluding paragraph at the end of the Discussion section to address this and a follow up studies will answer these questions.

Reviewer 2 Report

Comments and Suggestions for Authors

The present study demonstrated an interesting results on the diversity of folia fungal endophytes in grapevines from different resources, the authors also screened the endphyte bioactivity against the common grape pathogen Botrytis cinerea in vitro, analyzed the metabolite of bioactive strains. The manuscript is generally well written and easy to follow. I would like to see the author shown the results of the endophyte with biocontrol potential on diseased plant.

Introduction: I like the style the author started this part. The author may need more focus on the bioactivity of endophyte on pathogens, and also the potential mechanism of metabolite involved, this will improve the linkage of different components of the present study.

Materials and Methods: 2.6 Metabolite screening, the author did not tell the audience the target of metabolite screening, leaf of grapevine? Or the pathogen? Or the isolated endophyte fungi?

Results:

The diversity of endophyte isolated is much lower than I expected. Are the chemicals from the endophyte fungi have pathogen inhibition?

Discussion:

Afford information on the possible pathogen inhibition function of metabolites from the endophyte fungi.   

Author Response

The present study demonstrated an interesting results on the diversity of foliar fungal endophytes in grapevines from different resources, the authors also screened the endophyte bioactivity against the common grape pathogen Botrytis cinerea in vitro, analyzed the metabolite of bioactive strains. The manuscript is generally well written and easy to follow. I would like to see the author shown the results of the endophyte with biocontrol potential on diseased plant.

Response: Thanks for commenting on the scope of this work and your compliment about the manuscript, we appreciate that. Showing the effect of endophytes on biocontrol potential of disease on plants is the most interesting part of using endophytes as a biocontrol agent. One objective of this work was to do the pre-screening of endophytes on the media in dual culture competition assay to find out the most interesting candidates and to follow up them in planta assay for their biocontrol potential.  But doing these experiments needs quite some work to optimize the method for successful inoculation of endophytes on plants and then some molecular markers to follow them in plants and then challenging the plant with pathogens. We hope that this study will lead to the experiments of inoculating the most promising candidates from this study with the biocontrol potentials and then scoring the disease suppression potential of Botrytis and other common pathogens of grapevine. 

Introduction: I like the style the author started this part. The author may need more focus on the bioactivity of endophyte on pathogens, and also the potential mechanism of metabolite involved, this will improve the linkage of different components of the present study.

Response: Thanks for raising this question, we have added a few sentences in the introduction about the bioactivity of endophytes against pathogens and the potential mechanisms of endophytes to suppress pathogens.

Materials and Methods: 2.6 Metabolite screening, the author did not tell the audience the target of metabolite screening, leaf of grapevine? Or the pathogen? Or the isolated endophyte fungi?

Response: Thanks for point this out, sorry for missing this important information. We have added this now to the text in the section 2.6 of Materials and Methods. The metabolites were analyzed from fungal endophytes and Botrytis.

Results:

The diversity of endophyte isolated is much lower than I expected. Are the chemicals from the endophyte fungi have pathogen inhibition?

Response: Yes, we agree that diversity of the isolated endophytes is very low, but we will just like to mentioned that in this study we  focused exclusively on slow-growing fungal endophytes, i.e., those that took at least six days to emerge from the plated plant-tissue sample. Maybe we have missed some of the endophytes that grow faster.

Although we did not isolate the pure chemical compounds produced by endophytes against pathogens. But in dual culture competition assay, the endophytes are suppressing the pathogen growth from a distance, so we think the endophytes are secreting chemicals in the media that suppress the pathogen growth. 

Discussion:

Afford information on the possible pathogen inhibition function of metabolites from the endophyte fungi.  

Response: We have added few lines as suggested to the introduction and a conclusion paragraph at the end of the Discussion section to address these points.